# Normal Proteasome Function Is Needed to Prevent Kidney Graft Injury during Cold Storage Followed by Transplantation

**DOI:** 10.3390/ijms25042147

**Published:** 2024-02-10

**Authors:** Dinesh Bhattarai, Seong-Ok Lee, Lee Ann MacMillan-Crow, Nirmala Parajuli

**Affiliations:** 1Department of Pharmacology and Toxicology, University of Arkansas for Medical Sciences, Little Rock, AR 72205, USA; 2Division of Nephrology, University of Arkansas for Medical Sciences, Little Rock, AR 72205, USA

**Keywords:** cold storage and transplantation, proteasome subunits Rpt6 and B5, proteasome dysfunction, proteasome assembly chaperone PAC1, bortezomib

## Abstract

Kidney transplantation is the preferred treatment for end-stage kidney disease (ESKD). However, there is a shortage of transplantable kidneys, and donor organs can be damaged by necessary cold storage (CS). Although CS improves the viability of kidneys from deceased donors, prolonged CS negatively affects transplantation outcomes. Previously, we reported that renal proteasome function decreased after rat kidneys underwent CS followed by transplantation (CS + Tx). Here, we investigated the mechanism underlying proteasome dysfunction and the role of the proteasome in kidney graft outcome using a rat model of CS + Tx. We found that the key proteasome subunits β5, α3, and Rpt6 are modified, and proteasome assembly is impaired. Specifically, we detected the modification and aggregation of Rpt6 after CS + Tx, and Rpt6 modification was reversed when renal extracts were treated with protein phosphatases. CS + Tx kidneys also displayed increased levels of nitrotyrosine, an indicator of peroxynitrite (a reactive oxygen species, ROS), compared to sham. Because the Rpt6 subunit appeared to aggregate, we investigated the effect of CS + Tx-mediated ROS (peroxynitrite) generation on renal proteasome assembly and function. We treated NRK cells with exogenous peroxynitrite and evaluated PAC1 (proteasome assembly chaperone), Rpt6, and β5. Peroxynitrite induced a dose-dependent decrease in PAC1 and β5, but Rpt6 was not affected (protein level or modification). Finally, serum creatinine increased when we inhibited the proteasome in transplanted donor rat kidneys (without CS), recapitulating the effects of CS + Tx. These findings underscore the effects of CS + Tx on renal proteasome subunit dysregulation and also highlight the significance of proteasome activity in maintaining graft function following CS + Tx.

## 1. Introduction

The global occurrence of end-stage kidney disease (ESKD) is rapidly increasing, and there has been a 35% increase in ESKD over the last 10 years [1]. ESKD requires renal replacement therapy, such as dialysis or kidney transplantation, highlighting the significant burden of ESKD on the global healthcare system. Renal transplant is the preferred therapy, as dialysis is associated with several long-term risk factors, including infections, cardiovascular complications, and hypotension [2,3]. Moreover, compared to dialysis, renal transplantation is less expensive, increases long-term survival, and significantly improves quality of life [4,5].

At the same time, the shortage of organs on a global scale [5] and ischemia–reperfusion-mediated injury make kidney transplants clinically challenging [6,7]. Static cold storage (CS) is the established method for preserving renal grafts in most organ transplant centers. The CS process decelerates cellular activity and reduces the generation of harmful metabolites prior to transplantation [8,9]. Although CS preserves the quality of kidney grafts and reduces long-term post-transplant complications [10,11], prolonged CS is detrimental to long-term kidney transplant outcomes [12,13]. Data from experimental animal models further support the notion that prolonged CS activates several pathobiological pathways and compromises organ function after transplantation [14,15,16,17,18,19,20,21]. Therefore, it is important to understand the mechanisms of CS-mediated transplant injury to develop new therapeutics to protect kidney tissue during CS and improve graft function post-transplantation.

We previously reported that, in a rat model, CS increased mitochondrial reactive oxygen species (ROS), inactivated mitochondrial respiratory complexes, and decreased ATP levels [20,21,22]. Furthermore, we implanted CS kidneys in recipient rats (syngeneic model) and showed that CS combined with transplantation (CS + Tx) exacerbated mitochondrial dysfunction, which dysregulated renal protein homeostasis and kidney graft function [15,20,21,22]. Additionally, we showed that CS + Tx decreased proteasome function [15] in rat kidney grafts, but the role of proteasome and the mechanisms underlying its dysfunction during CS + Tx remain unknown.

The proteasome is a multimeric holoenzyme expressed in all eukaryotic cells where it recycles damaged or misfolded proteins via the ubiquitin–proteasome system [23,24]. Structurally, the proteasome consists of two primary domains: a 20S catalytic core particle, consisting of a stack of two β-rings and two α-rings, and one or two 19S regulatory particles, consisting of several regulatory subunits, including ATPase subunits [25,26,27]. The 20S β-ring consists of three to seven active sites, also known as β-catalytic subunits, that hydrolyze the peptide bonds in a caspase-like (β1 subunit), trypsin-like (β2 subunit), or chymotrypsin-like (β5 subunit) manner [28]. The six ATPase subunits (Rpt1-6) of a 19S regulatory particle are arranged in a ring that forms a gate for polyubiquitinated (Poly-Ub) protein substrates. When the ATPase ring recognizes a Poly-Ub protein substrate, the gate opens and allows the substrate to enter into the 20S catalytic core to be degraded into peptides [29,30,31].

The function or assembly of the mammalian proteasome can be influenced by the metabolic processes of cells. For example, reactive oxygen species (ROS) can impair proteasome assembly and reduce proteasome function [32,33]. Similarly, the O-GlcNAc-induced post-translational modification of the Rpt2 subunit of the mammalian proteasome by O-GlcNAcylation can reversibly decrease the proteolytic activity of the proteasome [34]. In contrast, the activated PKA-mediated phosphorylation of the Rpt6 subunit (Ser 120) can enhance the chymotrypsin-like peptidase activity of the proteasome [35]. Similarly, Rpt6 phosphorylation is essential for maintaining 26S proteasome assembly [36]. This implies that post-translational modifications to the ATPase subunits of the proteasome, particularly Rpt6, could significantly affect the assembly and function of the mammalian proteasome. Although proteasome activity decreases in the rat model of CS + Tx, the underlying mechanisms are not understood. 

Here, we hypothesized that CS-induced injury modifies proteasome subunits and impairs proteasome assembly in kidney grafts following transplantation. Using our established rat model of CS + Tx [15,16,17,20,21], we characterized the protein subunits and their assembly within the proteasome. We detected a decrease in the β5 subunit and an increase in modified Rpt6 subunits during CS + Tx. Importantly, this study underscores the critical role of these subunits in maintaining proteasome function, highlighting the importance of the proteasome for maintaining kidney graft function after transplantation. 

## 2. Results

### 2.1. Characterization of Proteasome Subunits during CS + Tx

Previously, we reported that the chymotrypsin-like (β5) peptidase activity of the proteasome decreased after CS + Tx when compared to sham or autotransplant (ATx) [15]. Here, we sought to investigate changes in the catalytic or regulatory subunits of the proteasome within the three experimental groups using kidney homogenates and denatured Western blots. First, we evaluated alterations in α- and β-ring subunits of the 20S proteasome in kidney homogenates after CS + Tx. The three β-catalytic subunits (β1, β2, and β5; β-ring) and the α3 subunit (α-ring) of the proteasome were evaluated in kidneys using Western blots. We observed a decreased level of β5 catalytic subunit in kidney tissues after CS + Tx, but β1- and β2-catalytic subunit levels remained unchanged (Figure 1A). Interestingly, we observed a unique pattern of α3 in a denatured Western blot (Figure 1B). The α3 subunit level at the predicted molecular weight remained unchanged across all groups (Figure 1B), but additional bands were detected at a higher molecular weight, and this phenomenon appeared in kidney homogenates only after CS + Tx (Figure 1B). Next, we characterized the levels of the 19S subunits of the proteasome. Because Rpt6 is an indispensable regulatory ATPase subunit of the proteasome and is crucial for proteasome assembly, we evaluated the effect of CS + Tx on the level of Rpt6. The predicted Rpt6 band (~49 kDa) remained unchanged in CS + Tx when compared to sham or ATx groups (Figure 1C). In addition to the predicted Rpt6 band, we detected bands with higher (~60 kDa) and lower (~25 kDa) molecular weight only after CS + Tx (Figure 1C). Next, we evaluated the levels of the Rpt5 subunit of the proteasome. The predicted Rpt5 band (~49 kDa) slightly decreased after CS + Tx when compared to sham or ATx groups (Figure 1C). Similar to the Rpt6 subunit, a lower-molecular-weight band of Rpt5 was detected, but the Western blots did not lead to the detection of a higher-molecular-weight band (Figure 1C). These results suggest that CS + Tx dysregulates 19S and 20S proteasome subunits in kidney grafts and that critical proteasome subunits, such as α3 and Rpt6, are subject to possible oxidative post-translational modification.

### 2.2. Native Proteasome Composition and Assembly after CS + Tx

To determine whether protein–protein interactions in the proteasome are affected by CS + Tx, we performed a nondenaturing (native) Western blot analysis of renal extracts using antibodies against Rpt6 (19S proteasome) and β5 (20S proteasome). β5 Western blot allowed us to detect the 30S-, 26S-, and 20S-proteasome complexes (mature), along with premature proteasome subcomplexes (SC). Compared to sham, β5 levels were unchanged within mature proteasome complexes (30S, 26S, and 20S) during CS + Tx, but premature complexes decreased after CS + Tx (Figure 2A, SC). Intriguingly, nonreducing Western blots for Rpt6 revealed a significant increase in abnormal proteasome complexes (26S and 19S) after CS + Tx (Figure 2B). Together with the reducing Rpt6 Western blot, these results suggest that the Rpt6 subunit aggregates during CS + Tx.

The proteasome is a multimeric enzyme that is assembled by proteasome assembly factors. PAC1 and PAC2 are mammalian proteasome assembly chaperones that initiate the assembly of α-ring formation that occurs during the initial assembly of the 20S proteasome [37,38,39], and in vitro data suggest that PAC1 knockdown decreases proteasome function [37,40,41]. To explore the effect of CS on proteasome assembly in kidney grafts, kidney extracts from transplant models were evaluated for PAC1. Western blots (reducing condition) showed no change in the PAC1 level in ATx kidneys when compared to the sham group (Figure 2C). Interestingly, PAC1 levels decreased in the kidneys after CS + Tx (Figure 2C), suggesting that a decrease in PAC1 might have impacted the assembly of the 20S proteasome pre-complex (Figure 2A).

### 2.3. PAC1 and β5 Are Sensitive to ROS during CS + Tx 

Oxidative stress is a hallmark of ischemia–reperfusion injury during CS + Tx. Nitrotyrosine, a footprint of the highly reactive oxidant, peroxynitrite, is a type of reactive oxygen species (ROS) and therefore can be used as a marker of oxidative stress. As expected, we observed an increase in nitrotyrosine immunostaining in kidney tissues after CS + Tx compared to sham kidneys (Figure 3A). Next, we sought to determine if oxidative stress dysregulated Rpt6 and β5 subunits during CS + Tx. We treated renal proximal tubular cells with peroxynitrite and observed a dose-dependent decrease in β5 (Figure 3B). Unexpectedly, Rpt6 protein levels remained unchanged after peroxynitrite treatment, and we did not observe higher-molecular-weight bands in the Rpt6 Western blot (Figure 3B), suggesting that the CS + Tx-derived phenotype of Rpt6 cannot be replicated through the bolus addition of peroxynitrite in NRK cells. Interestingly, peroxynitrite treatment decreased PAC1 levels (Figure 3B), suggesting that PAC1 is a target of oxidative stress.

### 2.4. Rpt6 Modification during CS + Tx Is Associated with Phosphorylation

It was intriguing to observe higher-molecular-weight bands of Rpt6 after CS + Tx, suggesting that Rpt6 was modified and/or aggregated (Figure 1); thus, we investigated the mechanism of Rpt6 modification. First, we hypothesized that this modification was due to glycosylation because O-linked N-acetylglucosaminylation (O-GlcNAcylation) via O-linked N-acetylglucosamine transferase (OGT) enzyme increases the molecular weight of a protein [35,42,43]. We used Western blot to evaluate the levels of OGT in kidney homogenates from sham, ATx, and CS + Tx groups. OGT decreased after CS + Tx, suggesting that Rpt6 was not modified via O-GlcNAcylation (Figure 4A). In parallel, we immunoprecipitated Rpt6 and subjected the immunoprecipitated product to mass spectrometry to evaluate methylation and oxidation. Unfortunately, neither protein modification was identified in the CS + Tx samples. Next, we considered whether this was a denaturation artifact (covalent) [44,45] of Rpt6 interacting with another protein/peptide [44,46,47]. For this, kidney RIPA lysates from sham, ATx, and CS + Tx groups were treated with increasing concentrations of dithiothreitol (DTT) for 30 min, but the higher-molecular-weight bands of Rpt6 were unchanged (Figure 4B).

Rpt6 phosphorylation is known to maintain the assembly and function of the proteasome [35,36]. Next, we evaluated Rpt6 subunit phosphorylation with SDS-PAGE Western blot and a commercial antibody for phosphorylated Rpt6 (Serine 120) [48]. The Western blots showed phosphorylated Rpt6 bands, along with the anticipated high-molecular-weight band of Rpt6 (Figure 5A). These data further suggest that Rpt6 modification is associated with phosphorylation. To confirm that Rpt6 modification is due to phosphorylation, kidney RIPA lysates were subjected to treatment using phosphatases (lambda and alkaline) to dephosphorylate serine, threonine, or tyrosine residues [36,48] followed by Western blot with Rpt6 antibody. Interestingly, phosphatase treatment eliminated the higher-molecular-weight band in the CS + Tx group (Figure 5B, arrow), suggesting that the aggregation was caused by phosphorylation. 

To examine the localization of phosphorylated Rpt6 in renal tissue, formalin-fixed renal sections were used for Rpt6 immunohistochemistry (phosphorylated and total). All three experimental groups (sham, ATx, and CS + Tx) exhibited a diffuse staining pattern of renal tubules for Rpt6, but there was a discrete staining pattern within the distal nephron for phosphorylated Rpt6 (Figure 5B). Notably, there was more phosphorylated Rpt6 in CS + Tx kidneys, particularly in proximal tubules, than in sham or ATx kidneys (Figure 5B, brown staining).

### 2.5. Proteasome Function Is Essential to Maintaining Renal Function following Transplantation

Our recent work with rat kidney transplant models suggests that chymotrypsin-like proteasome activity decreases in transplanted kidneys exposed to CS (CS + Tx) compared to those not exposed to CS (ATx) [15], and this impairs protein homeostasis [15,17]. We also reported that renal dysfunction was exacerbated in transplanted kidneys exposed to 4 or 18 h of CS (CS + Tx) [17,20,21], suggesting that renal dysfunction may be linked to proteasome dysfunction. Here, we sought to determine whether proteasome function is needed to maintain renal function during transplantation. For this, we inhibited proteasome function in donor rat kidneys (prior to transplantation) by flushing the kidneys with bortezomib (BTZ), a selective inhibitor of the β5 subunit of the proteasome [49]. The kidneys were then transplanted immediately (without CS). Kidney function analysis showed significantly increased serum creatinine levels in the BTZ + Tx group compared to sham or ATx. This level of kidney dysfunction was comparable to that of the 18 h CS + Tx groups (Figure 6A), suggesting that normal proteasome function is needed to maintain kidney function in the grafts. 

Finally, we sought to determine whether Rpt6 contributed to the chymotrypsin-like activity of the proteasome in renal tubule cells. Using RNA interference (siRNA), we knocked down β5 and Rpt6 subunits in rat kidney proximal tubular (NRK) cells (Figure 6B) and evaluated the chymotrypsin-like peptidase activity of the proteasome. As expected, the knockdown of β5 (catalytic subunit) decreased the chymotrypsin-like peptidase activity in NRK cells. However, a similar decline was observed following treatment with Rpt6 siRNA, which points to the strong role of Rpt6 in the observed CS + Tx-mediated loss of proteasome activity (Figure 6C).

## 3. Discussion

Our study revealed a few distinct characteristics of the renal proteasome following CS + Tx and further characterized the molecular events involved in CS-mediated proteasome dysfunction. A striking observation in this study was the CS-associated modification of α3 and Rpt6 subunits and the decrease in the β5 catalytic subunit. In this study, we focused on elucidating the causes of proteasome dysfunction and identified Rpt6 subunit modification and the decline in β5 levels as two distinct processes. It has been shown that the post-translational modification of Rpt6 by phosphorylation (S120) stabilizes the assembly of the proteasome and maintains its function [35,36]. Our data for the first time show a basal level of Rpt6 phosphorylation (S120) in rat kidneys. Notably, the phosphorylated Rpt6 appeared to generate a band shift in addition to the predicted molecular weight band, suggesting that Rpt6 is aggregated after CS + Tx. A caveat in this study was the difficulty in designing an in vitro model system of Rpt6 modification (phosphorylation/aggregation) recapitulating that from the in vivo CS + Tx model. Future studies are warranted to examine the mechanisms of aggregation and band shift of the phosphorylated Rpt6 during CS + Tx. 

In addition to phosphorylation, which increases proteasome activity, other post-translational modifications of the proteasome subunit include methylation, acetylation, carbonylation, O-GlcNAcylation, S-glutathionylation, myristoylation, and ubiquitination—all modifications that increase or decrease proteasome function (reviewed in [50]). Our CS + Tx model showed decreased levels of OGT in kidneys, suggesting that there is a lesser chance of Rpt6 modification via O-GlcNAcylation. Although the ubiquitination of proteasome subunits Rpn10 and Rpn13 were identified and shown to reduce substrate degradation by proteasome [51,52,53], future studies should examine post-translational modification, including ubiquitination, in the context of CS + Tx. Similarly, the cause of other subunits’ modification, such as α3, and its impact on proteasome dysfunction during CS + Tx has to be examined. 

Importantly, the levels of the β5 catalytic subunit decreased (denatured conditions) after CS + Tx. Although this decrease in β5 did not impact the levels of mature 20S, 26S, or 30S proteasome complexes in kidneys 1 day after transplantation (native conditions), the premature subcomplexes were disrupted after CS + Tx. Proteasome assembly requires the formation of an initiation complex with α subunits; then, β subunits join to form a subcomplex and eventually a mature complex of the proteasome [37,39,54]. The decrease in β5 subcomplexes (nonreducing conditions) only after CS + Tx suggests that CS promotes α3 subunit modification and impairs the assembly of the proteasome in kidney grafts after transplantation. Notably, *PAC1* is an essential gene, as complete knockout is embryonically lethal [55]. Here, we demonstrated that compared to sham or ATx, CS + Tx decreased PAC1, suggesting that this is a CS-mediated effect. In addition, we showed for the first time that ROS treatment reduced PAC1 levels in proximal tubular cells. PAC1 also functions to incorporate the β subunits (including β5) into the proteasome initiation complex in association with another assembly chaperone, hUmp1 [37], and our in vitro assay showed a decrease in β5 in proximal tubular (NRK) cells following ROS exposure. On the other hand, β5 inhibition induces mitochondrial ROS in proximal tubular cells [56], and mitochondrial dysfunction further contributes to the decline in proteasome function [15]. Taken together, we suggest that CS + Tx-mediated oxidative stress decreases PAC1 and β5 subunits in kidney grafts, impairing the assembly and function of the proteasome. 

Additionally, we highlighted, for the first time, the important role of normal proteasome activity on kidney function following transplantation. This was shown by treating donor rat kidneys prior to transplantation (no CS) with the proteasome inhibitor bortezomib (BTZ), which resulted in declined renal function after transplantation. These data suggest that normal proteasome function is needed during CS to prevent renal injury in transplants. With BTZ, we defined the role of the β5 subunit in kidneys after transplantation. Importantly, by applying BTZ to the donor kidney before transplantation, we minimized the systemic circulation of the drug, avoiding the potential side effects of inhibiting the proteasome in other tissues. Moreover, our RNA interference data demonstrate that Rpt6 is essential for chymotrypsin-like peptidase activity in proximal tubular cells, suggesting that CS-mediated Rpt6 modification contributes to proteasome dysfunction within the renal tubules.

## 4. Materials and Methods

### 4.1. Animals

Male Lewis rats (200–250 g) were purchased from Charles River Laboratories and used as kidney transplant donors and recipients. All animal protocols were approved by the Institutional Animal Care and Use Committee at the University of Arkansas for Medical Sciences, and all animal experiments were performed in compliance with institutional and NIH guidelines.

### 4.2. Rat Surgery and Experimental Groups

Rat surgery was performed as described previously [15,20,21]. Isoflurane anesthesia was used during the surgery. 

#### 4.2.1. Rat Kidney Cold Storage and Transplantation (CS + Tx) Surgery (Donor Kidney)

The right and left kidneys were removed from donor rats and flushed with and stored in a cold (4 °C) University of Wisconsin solution (hereafter referred to as CS solution) for 4–18 h. The right donor kidney was saved immediately as the CS kidney, and the left donor kidney was transplanted into a recipient rat (CS + Tx). In the recipient rat, the native left kidney was removed, and the donor left kidney (4- or 18 h CS) was transplanted via the end-to-end anastomosis of the recipient and donor renal vessels (artery and vein). The ureter was anastomosed end to end over a 5 mm PE-10 polyethylene stent. The native right kidney was removed prior to the muscle/skin closure of the abdomen so that the function of the graft kidney could be isolated. Postoperatively, animals were given 0.9% (*w*/*v*) NaCl (saline solution) subcutaneously (SC) in the dorsal region and placed on a heating pad to recover from anesthesia. The rats were given buprenorphine (2 mg/kg, SC) for pain. After 1 day of reperfusion, the transplanted left kidney and blood were collected under anesthesia and saved as the CS + Tx group (n = 9). 

#### 4.2.2. Autotransplant (ATx) Surgery

The ATx surgery was included so that the effect of CS could be isolated from the effect of transplant surgery alone. ATx surgery (n = 9) was performed as described for the CS + Tx group above, except that both kidneys were removed, and the left kidney was flushed with warm saline and immediately transplanted back into the same rat without CS exposure. After 1 day of reperfusion, the transplanted left kidney and blood were collected under anesthesia and saved as the CS + Tx group (n = 9). This group served as a transplant control for the CS + Tx group. 

#### 4.2.3. Sham Surgery

Rats underwent the same procedure for right nephrectomy but without renal transplantation (sham operation). The right kidney was saved immediately as an untreated control kidney. The left kidney (sham) and blood were harvested 1 day post-surgery. The sham group (n = 9) served as a healthy control group. 

### 4.3. Tissue Sample Collection

Kidneys and blood (with or without heparin) were collected under anesthesia 1 day post-surgery, and the rats were euthanatized by exsanguination as described in the approved animal-use protocol. Kidneys were immediately flash-frozen and saved at −80 °C until further use. For blood processing, blood without heparin was allowed to clot on ice for 1 h. The blood (with or without heparin) was then centrifuged (5000× *g*) at 4 °C for 10 min to remove the clot, and the supernatant (serum or plasma) was aliquoted and saved at −80 °C until further use.

### 4.4. In Vitro Cell Model

Normal rat kidney proximal tubular cell cultures (NRK-52E, American Type Culture Collection, Manassas, VA, USA) were maintained in a warm (37 °C) growth medium (DMEM plus 5% fetal calf serum and 1% penicillin/streptomycin). NRK cells at 70% confluence were washed with warm PBS (37 °C) and treated with peroxynitrite by adding peroxynitrite (0, 30, and 300 µM, Calbiochem) to PBS on the plate at 37 °C for 10 min. PBS was then removed and replaced with an NRK cell growth medium, and the cells were returned to the incubator at 37 °C for 18 h. The control NRK cells were treated with vehicle (PBS) only. After treatment, NRK cells were washed 2 times with cold PBS (4 °C) and used to prepare RIPA lysates or cell extracts for proteasome function studies [15,16,57]. 

### 4.5. Renal Function and Serum Chemistry: iSTAT Measurement

Blood chemistry was assayed in heparinized blood (arterial) using an iSTAT^TM^ hand-held clinical chemistry analyzer and CHEM8^+^ cartridges as described by the manufacturer (Vetscan^®®^, Abaxis, Union City, CA, USA) [20,21].

### 4.6. Proteasome Function

The peptidase activity of the β5 subunit was measured in NRK cells by the hydrolysis of a fluorogenic peptide substrate specific for β5 [15]. Kidney tissue extracts were prepared with proteasome extraction buffer [15] and incubated with the peptide substrate, acetyl-Ala-Asn-Trp-AMC (100 µM) (South Bay Bio, San Jose, CA, USA) in the presence or absence of the proteasome inhibitor ONX 0914 (10 µM) (Cayman Chemical, Ann Arbor, MI, USA). After 30 min incubation at 37 °C, fluorescence was measured (excitation, 380 nm; emission, 460 nm) with a BioTek Synergy H1 plate reader (Agilent, Santa Clara, CA, USA) [15]. 

### 4.7. SDS-PAGE/Native PAGE and Immunoblotting

For SDS-PAGE Western blotting, renal extracts from whole-kidney homogenates were prepared with RIPA lysis buffer (Pierce, Rockford, IL, USA) (reducing condition) [58]. Renal extracts (20 µg cells and 30 µg tissues) were separated with SDS-PAGE and transferred to a PVDF membrane. After transfer, the membrane was incubated with antibodies against β1 (1:1000; Enzo, Farmingdale, NY, USA), β2 (1:1000; Enzo), β5 (1:1000; Abcam, Waltham, MA, USA), α3 (1:1000; Abcam), Rpt6 (1:1000; Enzo), Rpt5 (1:1000; Enzo), PAC1 (1:1000; Abcam), OGT (1:1000; Abcam), and β-actin (loading control, 1:1000; Sigma, St. Louis, MO, USA). Probed membranes were washed 3 times, incubated with horseradish peroxidase-conjugated secondary goat anti-mouse or goat anti-rabbit antibodies (1:30,000; Seracare KPL, Milford, MA, USA), and assayed for enhanced chemiluminescence (Thermo Fisher Scientific, Waltham, MA, USA). Densitometry was performed with AlphaEase FC software v6.0 (Alpha Innotech, San Leandro, CA, USA).

For native PAGE and immunoblotting, renal extracts from whole-kidney homogenates were prepared with 0.9% digitonin lysis buffer as described in [15]. Renal extracts (20 µg) were resolved with a Bis-Tris (4–12%) gel and transferred to a PVDF membrane [59]. After transfer, the membrane was incubated with 1X Red Alert Western Blot Stain (Millipore Sigma, Burlington, MA, USA) for 10 min at room temperature. The membrane was quickly washed with ddH_2_O to visualize protein bands, and images were taken with a FluorchemTM 8900. Western blot analysis was performed with antibodies for β5 (1:1000; Abcam) and Rpt6 (1:1000; Enzo). Chemiluminescence-based detection and densitometry were performed as described above. For all native gel Western blot analyses, a densitometry ratio of β5 to the corresponding whole-lane density of Red Alert^TM^ stain was considered for statistical evaluation [16].

### 4.8. Protein Dephosphorylation Assay

For the protein dephosphorylation assay, the phosphatase treatment of the RIPA extract followed by SDS-PAGE Western blotting was performed using lambda and alkaline phosphatases (Sigma). Briefly, 30 µg of RIPA lysate (without phosphatase inhibitor) was aliquoted (2 aliquots per sample), and phosphatase assay buffer (provided in the kit) was added to the lysate to make a final concentration of 1 µg/µL. In one aliquot, lambda and alkaline phosphatases (0.5 uL each) were added (+phosphatase), and an equal volume of assay buffer was added to another lysate aliquot (−phosphatase). The lysates (+/−phosphatase) were then incubated at 37 °C for 30 min. The phosphatase reaction was stopped by boiling the samples in Laemmli SDS buffer. Renal extracts (+/−phosphatase) were employed for SDS-PAGE and Western blotting as described above using Rpt 6 (1:1000; Enzo) and phospho-Rpt6 antibodies (S120; 1:1000; ProSci, Poway, CA, USA).

### 4.9. Immunohistochemistry

Kidneys were formalin-fixed and paraffin-embedded. Two cross-sections (4–5 µm thickness) from each paraffin block of kidneys were mounted on a glass slide. The sections were deparaffinized with xylene and hydrated through a graded series of ethanol. For immunohistochemical analysis, antigens were retrieved by heating sections in 10 mM of sodium citrate buffer (pH 6.0) for 30 min. Endogenous peroxidase was quenched by incubating the sections with a BLOXALL^TM^ Endogenous Peroxidase and Alkaline Phosphatase Blocking Solution (Vector) for 15 min at room temperature, and the sections were blocked with blocking solution (2.5% normal goat serum) for 20 min at room temperature. Primary antibodies were diluted in an antibody diluent solution (1% BSA and 0.5% nonfat dry milk in Tris-buffered saline) and incubated overnight at 4 °C. The diluted primary antibodies were as follows: anti-Rpt6 (1:500; Enzo Life Sciences), anti-phospho-Rpt6 (1:500; ProSci), and anti-nitrotyrosine (1:3000; Millipore Sigma). Immunoreactivity was detected with ImmPRESS^TM^ Reagent Anti-Rabbit IgG (for rabbit primary antibody) or Anti-Mouse IgG (for mouse primary antibody) (Vector) and an ImmPACT^TM^ DAB Peroxidase Substrate. All images were taken using a Nikon Eclipse E800 microscope with Nikon Elements software v5.21. 

### 4.10. Statistical Analysis

Data are presented as means ± standard error of the mean (SEM) (GraphPad Prism, Version 9). Data (n = 3–9) were analyzed with a one-way ANOVA and Tukey’s post hoc test for multiple-group comparisons, and an unpaired Student’s *t*-test was used to detect differences between the means of two groups at a 95% confidence level. Differences with *p* < 0.05 were considered statistically significant. In group comparisons, the Sham group served as controls for both transplant models (ATx and CS + Tx) because all underwent a right nephrectomy (removal of right kidney). CS + Tx groups were compared with ATx to evaluate the effects of CS on transplantation. 

## 5. Conclusions

In summary, we identified CS-specific alterations to proteasome subunits ꞵ5, α3, and Rpt6. Specifically, ꞵ5 decreased, and Rpt6 was modified after CS + Tx, and this resulted in decreased proteasome function (chymotrypsin-like). RNA interference demonstrated that Rpt6 is essential for chymotrypsin-like peptidase activity within the renal tubules, suggesting that Rpt6 modification contributed to the observed decline in proteasome function after CS + Tx [15]. We also observed a decrease in proteasome assembly chaperone PAC1 and further showed that this decrease depended on oxidative stress. Although Rpt6 protein modification was unprecedented and related to phosphorylation, we could not pinpoint the specific mechanisms of Rpt6 modification and its band shift as detected by Western blotting. Importantly, this study revealed a novel association between proteasome activity and renal function, such that functional proteasomes are needed to reduce renal injury during transplantation. Future studies should target assembly factors, such as PAC1, or work to establish a functional proteasome during CS + Tx toward mitigating renal graft injury. As organ transplantation continues to be a critical intervention for ESKD, it is essential to understand the molecular pathways that contribute to graft injury/dysfunction to develop strategies to improve transplant success rates and long-term patient outcomes. The current study highlights the essential role of proteasome function in maintaining kidney function and suggests that prolonged CS reduces proteasome function by impairing its structural modification (subunit alteration) in transplants. Future studies designed to mitigate proteasome dysfunction during CS + Tx may improve graft outcomes to benefit patients with ESKD.

## Figures and Tables

**Figure 1 ijms-25-02147-f001:**
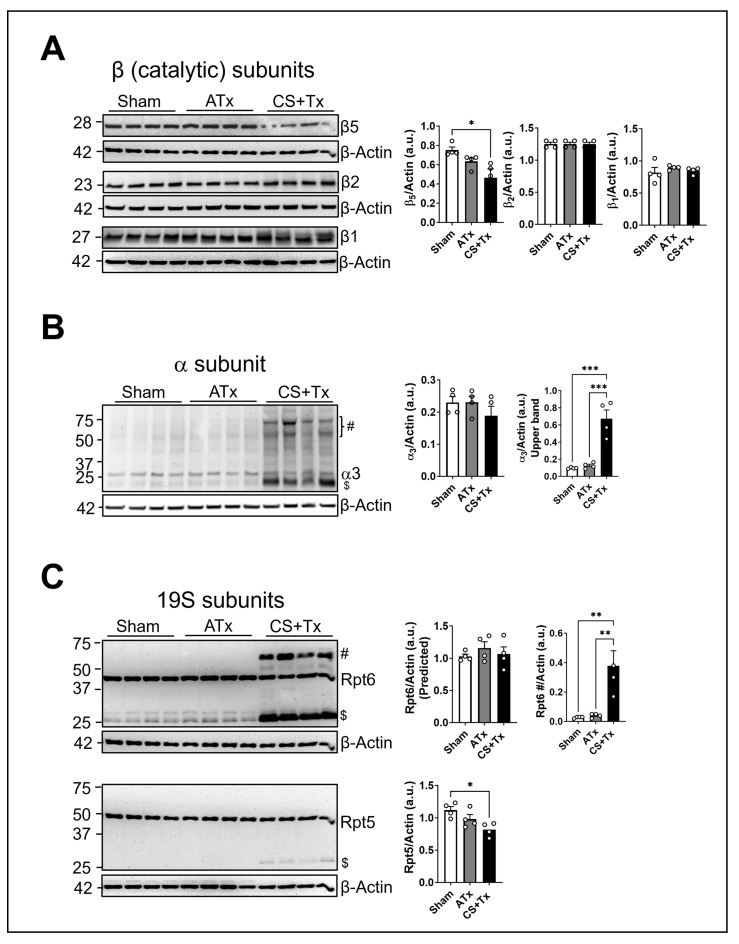
Cold storage dysregulates proteasome subunits in rat kidney grafts after transplantation. Lewis rat kidneys were flushed with and stored in cold storage (CS) solution (4 °C) for 18 h followed by transplantation to a recipient Lewis rat (n = 4/group). Autotransplant (ATx, transplant with no CS) rats were used as transplant control, and sham rats (right nephrectomy, control kidney) were used as healthy controls: (**A**) Immunoblot of catalytic subunits of the proteasome (β1, β2, and β5) in rat kidney homogenates from sham, ATx, and CS followed by transplant (CS + Tx) groups (n = 4/group). (**B**) Immunoblot of the α-regulatory subunit of the proteasome (α3) in rat kidney homogenates from sham, ATx, and CS + Tx groups (n = 4/group); # indicates modified α3 (~50–70 kDa), and $ indicates cleaved α3 (~22 kDa) protein. (**C**) Immunoblot of ATPase regulatory subunits of the proteasome (Rpt6 and Rpt5) in rat kidney homogenates from sham, ATx, and CS + Tx groups (n = 4/group); # indicates modified Rpt6 subunit (~60 KDa), and $ indicates cleaved Rpt5 or Rpt6 protein. β-Actin served as a loading control. For all Western blots, representative blots from 3 independent experiments are shown. Graphs show densitometry (mean ± SEM) normalized to β-actin (n = 4/group); *p* < 0.05 was considered statistically significant. * *p* < 0.05; ** *p* < 0.01; *** *p* < 0.001.

**Figure 2 ijms-25-02147-f002:**
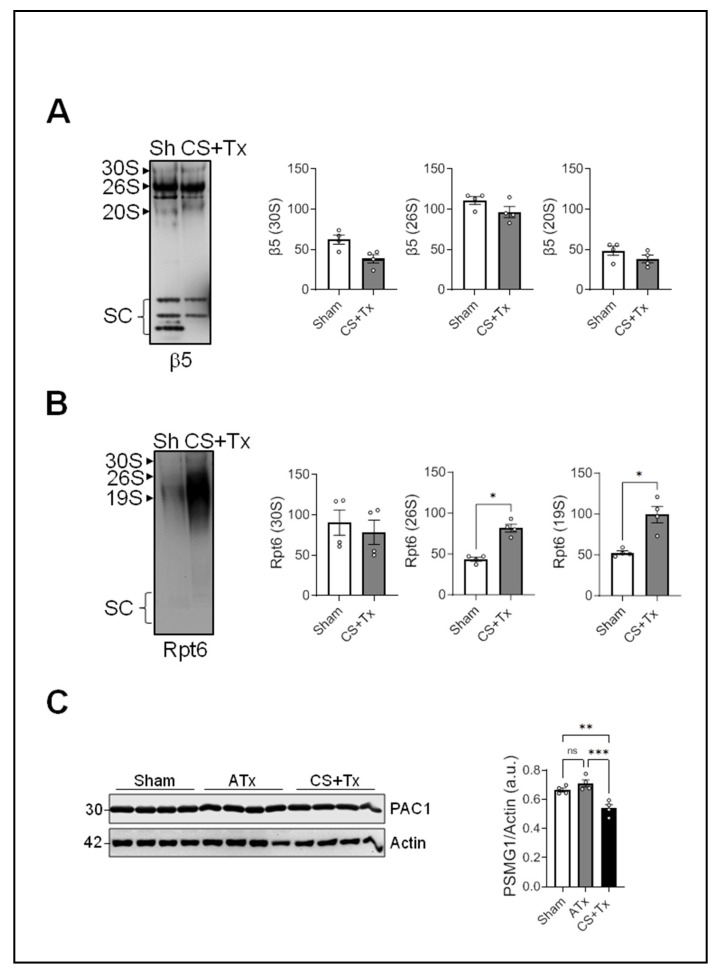
CS + Tx disrupts native proteasome in rat kidney grafts. Lewis rat kidneys were flushed with and stored in cold storage (CS) solution (4 °C) for 4 h or 18 h followed by transplantation to a recipient Lewis rat. Sham or autotransplant (ATx) rats were used as controls: (**A**,**B**) Native immunoblot of β5 (**A**) and Rpt6 (**B**) in rat kidney homogenates from sham and cold storage and transplant (CS + Tx) groups (n = 4/group). Representative blots from 3 independent experiments are shown. SC = subcomplex of the proteasome (immature). Graphs show densitometry (mean ± SEM) normalized to GAPDH (n = 4/group). (**C**) SDS-PAGE Western blot for PAC1. Representative blots from 3 independent experiments are shown for (**A**–**C**). Bar graphs show densitometry (mean ± SEM) normalized to β-actin (n = 4/group); *p* < 0.05 was considered statistically significant. * *p* < 0.05; ** *p* < 0.01; *** *p* < 0.001; ns: not significant.

**Figure 3 ijms-25-02147-f003:**
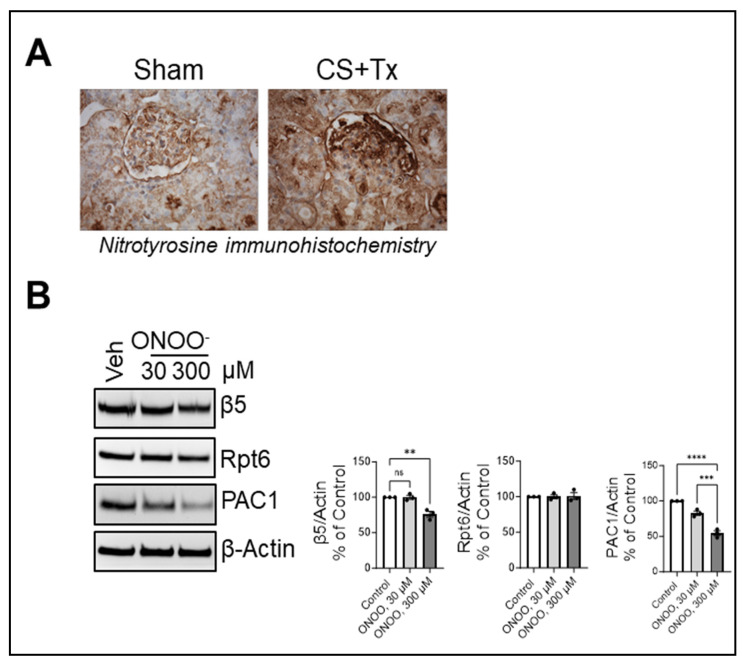
Reactive oxygen species dysregulate proteasome subunits in kidney grafts after cold storage plus transplantation. Lewis rat kidneys (n = 3/group) were flushed with and stored in cold storage (CS) solution (4 °C) for 18 h followed by transplantation to a recipient Lewis rat. Sham was used as a control: (**A**) Kidney tissues (paraffin-embedded section) from sham and CS + Tx groups were subjected to immunohistochemistry with anti-nitrotyrosine antibody. Brown staining indicates reactivity to nitrotyrosine. (**B**) Proteins from RIPA renal extracts (30 µg) prepared from NRK cells after peroxynitrite treatment (ONOO^−^, 0–300 µM) were subjected to SDS-PAGE and Western blot for Rpt6, β5 and PAC1. β-Actin was used as a loading control. A representative blot from 3 independent experiments is shown. Differences between group means were compared with ANOVA (>3-group comparison); *p* < 0.05 was considered statistically significant. ** *p* < 0.01; *** *p* < 0.001, **** *p* < 0.0001, ns: not significant.

**Figure 4 ijms-25-02147-f004:**
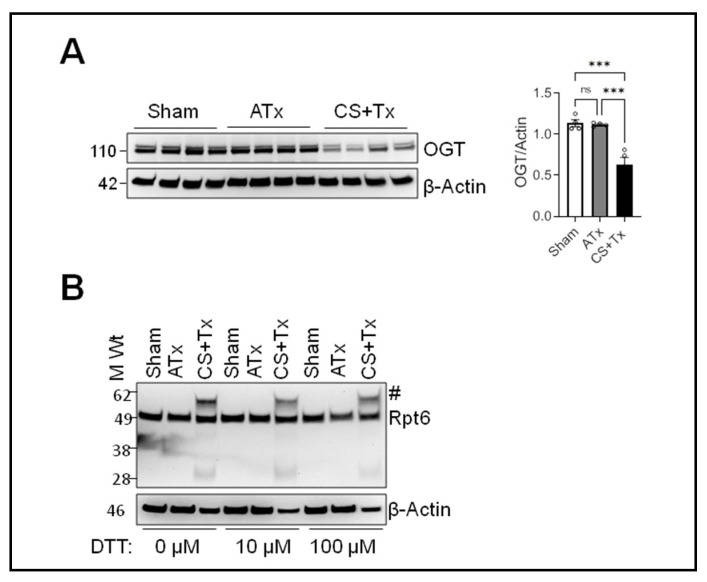
Modification of Rpt6 proteasome subunit in kidney grafts after cold storage plus transplantation is not related to glycosylation or impaired disulfide bond formation. Lewis rat kidneys were flushed with and stored in cold storage (CS) solution (4 °C) for 4 h or 18 h followed by transplantation to a recipient Lewis rat. Sham or ATx rats were used as controls: (**A**) Immunoblot of O-linked GlcNAcylation transferase (OGT) protein in rat kidney homogenates from sham, autotransplant (ATx), and cold storage followed by transplant (CS + Tx) groups (n = 4/group). β-Actin was used as a loading control. A representative blot from 3 independent experiments is shown. Differences between group means were compared with ANOVA (3-group comparison); *p* < 0.05 was considered statistically significant. *** *p* < 0.001. (**B**) Proteins from RIPA renal lysates (30 µg) prepared with increasing concentrations of dithiothreitol (DTT) (0–100 µM) from sham, ATx, and CS + Tx groups were subjected to SDS-PAGE and Western blot for Rpt6. β-Actin was used as a loading control. A representative blot from 3 independent experiments is shown. ns: not significant.

**Figure 5 ijms-25-02147-f005:**
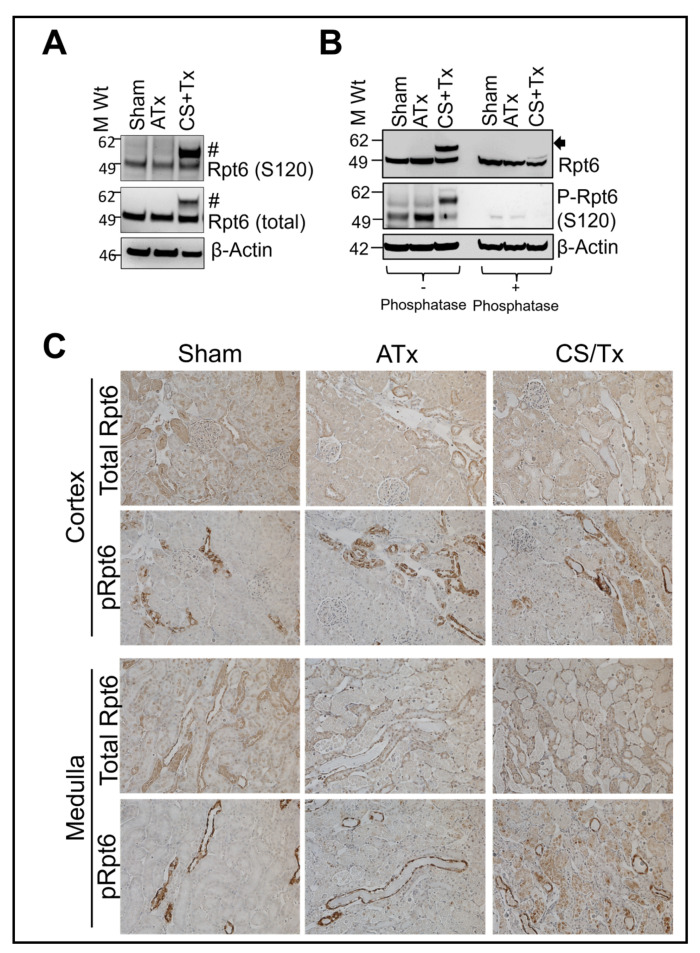
Rpt6 modification is associated with phosphorylation in kidney grafts after cold storage plus transplantation. Lewis rat kidneys (n = 3/group) were flushed with and stored in cold storage (CS) solution (4 °C) for 4 h or 18 h followed by transplantation to a recipient Lewis rat. Sham or ATx rats were used as controls: (**A**) Proteins from RIPA renal lysates (30 µg) from sham, ATx, and CS + Tx groups were subjected to SDS-PAGE and Western blot for phosphorylated (S120) Rpt6 (#) and total Rpt6. β-Actin was used as a loading control. A representative blot from 3 independent experiments is shown. (**B**) RIPA renal lysates (30 µg) treated with or without phosphatase from sham, ATx, and CS + Tx groups were subjected to SDS-PAGE and Western blot for phosphorylated (S120) and total Rpt6. β-Actin was used as a loading control. A representative blot from 3 independent experiments is shown. Arrow indicates modified Rpt6. (**C**) Kidney tissues (paraffin-embedded section) from sham and CS + Tx groups were subjected to immunohistochemistry with anti-Rpt6 or anti-phospho-Rpt6 (S120) antibodies. Brown staining indicates reactivity to Rpt6 or phospho-Rpt6 (S120) in the cortical and medullary regions. Representative images of n = 3/group are shown.

**Figure 6 ijms-25-02147-f006:**
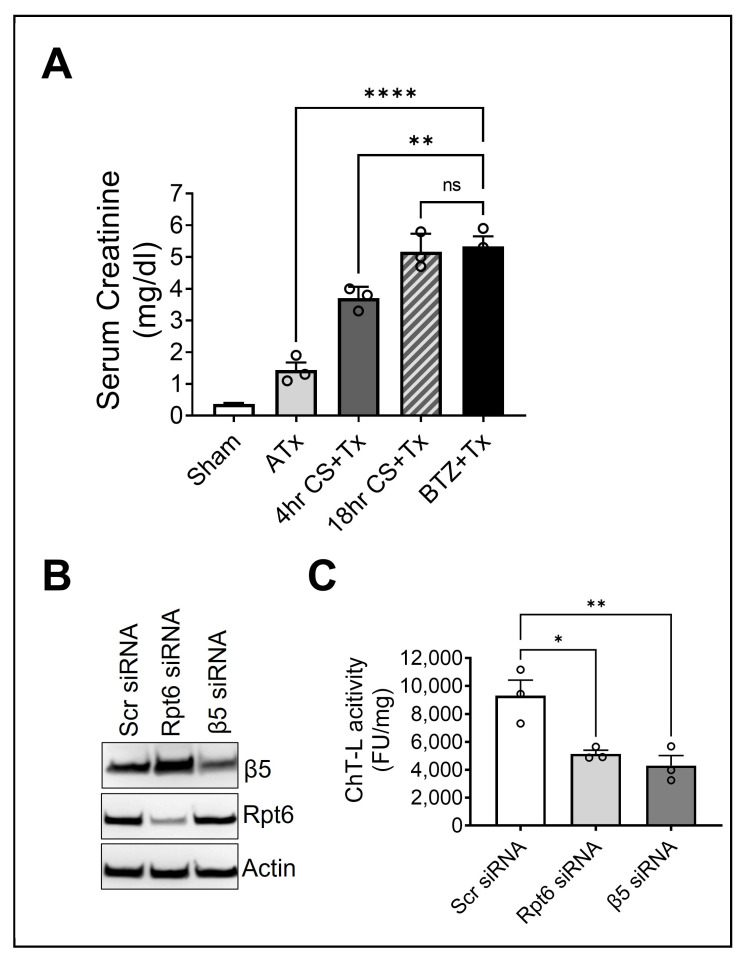
Treating donor rat kidneys with proteasome inhibitor (without CS) before transplantation impairs graft function: (**A**) Lewis rat kidneys were flushed with bortezomib (proteasome inhibitor) (without cold storage [CS]) followed by immediate transplantation to a recipient Lewis rat. Sham, ATx, or CS + Tx (4- or 18 h CS) rats were used as controls. Renal function was assayed in rat blood by measuring serum creatinine. Data are the mean ± SEM (bar graphs, n = 3/group). Differences between group means were compared with Student’s *t*-test (2-group comparison) or ANOVA (3-group comparison). (**B**) Representative immunoblots (n = 3) showing protein knockdown via siRNA in rat proximal tubular (NRK) cells after transfection with Rpt6 or β5 siRNA (50 nM). Nontargeting (scrambled [Scr] siRNA) was used as a control. (**C**) Chymotrypsin-like proteasome function (β5 peptidase activity) assayed in NRK cell extracts using a peptide substrate (Suc-LLVY-AMC). Data are the mean ± SEM (bar graph, n = 3/group); *p* < 0.05 was considered statistically significant. * *p* < 0.05, ** *p* < 0.01 and **** *p* < 0.0001. ns: not significant.

## Data Availability

All data associated with this study are present in the paper. Data will be made available upon reasonable request.

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
