# Peer review of "Normal Proteasome Function Is Needed to Prevent Kidney Graft Injury during Cold Storage Followed by Transplantation"

_ijms, 2024, doi:10.3390/ijms25042147_

Round 1

Reviewer 1 Report

Comments and Suggestions for Authors

This is an interesting article about an important topic of kidney transplant graft performance after cold storage. 

Minor issues:

  • The figure quality could be improved, especially the axes labels seem blurred (this is likely a problem from converting)

  • Total Rpt6 staining is not convincing; it appears non-present/negative control should be provided. The abbreviation left label should be corrected - currently, it is  Rpt6 vs pRPT6.

Author Response

This is an interesting article about an important topic of kidney transplant graft performance after cold storage. 

Comment: Thank you for appreciating the work.

Comment 1: The figure quality could be improved, especially the axes labels seem blurred (this is likely a problem from converting)

Response: We apologize for the inconvenience. We have fixed it now.

Comment 2: Total Rpt6 staining is not convincing; it appears non-present/negative control should be provided. The abbreviation left label should be corrected - currently, it is  Rpt6 vs pRPT6.

Response: Thank you for your suggestions. In fact, we have performed isotype-matched control for both Rpt6 (mouse IgG) and pRpt6 (rabbit IgG) antibodies. The isotype-matched control antibodies do not pick brown staining. The data is presented for review purposes only (see below). We also have fixed the abbreviation for phosphor-Rpt6.

Reviewer 2 Report

Comments and Suggestions for Authors The manuscript sheds light on the mechanism underlying proteasome dysfunction and its role in kidney graft outcome, providing valuable insights for improving transplantation outcomes.   In my opinion, this manuscript is well planned, executed and written, and also contains bibliographical references that support what they are trying to explain.
The only thing I could contribute is the following:

- It would be useful to indicate the number of samples analysed for each result, since according to the "statistical analysis" section it is stated to be 3-9, but it is not specified for each graph. This is important for the correct interpretation of the results.
- What has been the survival rate of the transplanted animals?
- Regarding the Western blots, was a single gel made for each experiment or is it a representative image? Were triplicates of each gel made to check the reliability of the results?   - I would recommend including a figure explaining the experimental groups (4.2.1) to make it easier to understand how the transplantation was performed in each group.   - Although it is referenced in the text, it would be appropriate to highlight: What are the possible clinical implications of this study?

The manuscript presents a well-planned novel study that scientifically answers the study questions. The writing is correct. In my opinion, it can be published.

Author Response

The manuscript sheds light on the mechanism underlying proteasome dysfunction and its role in kidney graft outcome, providing valuable insights for improving transplantation outcomes.   In my opinion, this manuscript is well planned, executed and written, and also contains bibliographical references that support what they are trying to explain.

Comment: Thank you for appreciating the work.

The only thing I could contribute is the following:

Comment 1: It would be useful to indicate the number of samples analysed for each result, since according to the "statistical analysis" section it is stated to be 3-9, but it is not specified for each graph. This is important for the correct interpretation of the results.

Response: We have indicated this for each figure (refer to Figure Legends).

Comment 2: What has been the survival rate of the transplanted animals?

Response: We have performed survival studies up to 7 days post-surgery. Our data shows that the survival rate 1-day post-transplantation (with 18h CS, both native kidneys removed) is > 98%. However, rats won’t survive beyond 3 days post-transplant (18h CS, both native kidneys removed). Whereas, the survival rate is 100% for the rats with 4h CS followed by transplantation (for up to 7 days post-transplant study). And, rats have no issues with survival under the condition where the right native kidney remains post-transplant. These data are included in our manuscript, which was recently accepted for publication (Bhattarai et al., 2024; Accepted Kidney360; Assigned DOI: 10.34067/KID.0000000000000368).

Comment 3: Regarding the Western blots, was a single gel made for each experiment or is it a representative image? Were triplicates of each gel made to check the reliability of the results?

Response: For all western blot experiments, SDS-PAGE was run/repeated more than 2 times and the figures represent a representative image. Only reliable results are presented in this manuscript. In most instances, the bar graph represents the average per group from the same western blot. This reduces the variability issue coming from different blots with different densitometry intensities.  

Comment 4: I would recommend including a figure explaining the experimental groups (4.2.1) to make it easier to understand how the transplantation was performed in each group.  

Response: Sorry for not making it clearer. We have performed surgery with a sample size of N=9 per group. However, when we perform assays, we start with a small sample size (N) for evaluation until we get statistically significant data, and we try to minimize the use of rat samples complying with NIH’s  3R policy (replacement, reduction, and refinement). Because kidney transplants are very invasive and the results are profound, and in most cases, data are significant with smaller N, e.g., N=3).

For the manuscript, the sample size for each assay/experiment is given under the figure legend. Additionally, we have present graph bars with dots, and each dot represents an animal within that group.

Comment 5: Although it is referenced in the text, it would be appropriate to highlight: What are the possible clinical implications of this study?

Response: Thank you for the suggestions. We have now included this section under the Discussion.

Comment 6: The manuscript presents a well-planned novel study that scientifically answers the study questions. The writing is correct. In my opinion, it can be published.

Response: Thank you for liking the work. We appreciate it.
